# Differential Regulation of Two Arms of mTORC1 Pathway Fine-Tunes Global Protein Synthesis in Resting B Lymphocytes

**DOI:** 10.3390/ijms232416017

**Published:** 2022-12-16

**Authors:** Gagan Dev, Amanpreet Singh Chawla, Suman Gupta, Vineeta Bal, Anna George, Satyajit Rath, G. Aneeshkumar Arimbasseri

**Affiliations:** 1National Institute of Immunology, Aruna Asaf Ali Marg, New Delhi 110067, Delhi, India; 2MRC Protein Phosphorylation and Ubiquitylation Unit, School of Life Sciences, University of Dundee, Dundee DD1 5EH, UK; 3Department of Medicine, Cedars-Sinai Medical Center, Los Angeles, CA 90048, USA; 4Indian Institute of Science Education and Research, Dr. Homi Bhabha Rd., Pashan, Pune 411008, Maharashtra, India; 5Translational Health Sciences and Technology Institute, NCR Biotech Science Cluster, Faridabad 121001, Haryana, India

**Keywords:** protein synthesis, mTORC1, 4EBP1, Ribo-Seq, B cell, T cell

## Abstract

Protein synthesis is tightly regulated by both gene-specific and global mechanisms to match the metabolic and proliferative demands of the cell. While the regulation of global protein synthesis in response to mitogen or stress signals is relatively well understood in multiple experimental systems, how different cell types fine-tune their basal protein synthesis rate is not known. In a previous study, we showed that resting B and T lymphocytes exhibit dramatic differences in their metabolic profile, with implications for their post-activation function. Here, we show that resting B cells, despite being quiescent, exhibit increased protein synthesis in vivo as well as ex vivo. The increased protein synthesis in B cells is driven by mTORC1, which exhibits an intermediate level of activation in these cells when compared with resting T cells and activated B cells. A comparative analysis of the transcriptome and translatome of these cells indicates that the genes encoding the MHC Class II molecules and their chaperone CD74 are highly translated in B cells. These data suggest that the translatome of B cells shows enrichment for genes associated with antigen processing and presentation. Even though the B cells exhibit higher mTORC1 levels, they prevent the translational activation of TOP mRNAs, which are mostly constituted by ribosomal proteins and other translation factors, by upregulating 4EBP1 levels. This mechanism may keep the protein synthesis machinery under check while enabling higher levels of translation in B cells.

## 1. Introduction

Protein synthesis is one of the most energy-consuming processes in a cell and is tightly regulated in response to environmental conditions [1]. Several external and internal stimuli and pathways upregulate protein synthesis in eukaryotic cells [2,3]. Most of these pathways converge on the mTORC1 complex, which phosphorylates several regulatory factors of protein synthesis machinery [4,5,6]. The mTORC1 pathway upregulates protein synthesis primarily through the inhibitory phosphorylation of 4EBP proteins. The 4EBPs bind to eIF4E and thereby inhibit cap-dependent translation; their phosphorylation by mTORC1 derepresses protein synthesis by inhibiting 4EBP activity [2,7]. Apart from this de-repression activity of mTORC1, P70-S6 kinase, another target of mTORC1, phosphorylates other translation initiation factors and ribosomal protein S6 to activate translation [8].

The specific roles of different arms (4EBP and p70S6K) of the mTORC1 pathway have been studied in detail. In fibroblasts, the deletion of 4EBPs affected the regulation of cell proliferation, not the growth [9]. In skeletal muscles, the deletion of S6K1 led to a decrease in myoblast growth without affecting their proliferation [10]. Rapid cell growth and proliferation are hallmarks of activated lymphocytes, requiring immediate upregulation of total protein synthesis after activation. Different phases of the immune response, including proliferation and differentiation, are marked by distinct changes in the protein synthetic capacity [11]. Moreover, in lymphocytes, after activation, 4EBPs are important for both cell growth and proliferation, indicating that the functional division of these pathways varies with the cell type [12].

The major targets of mTORC1 at the translation level are the terminal oligopyrimidine-containing (TOP) mRNAs [13,14]. This class of mRNAs mostly consists of ribosomal protein-coding genes and translation factors [13], and their regulation will have an impact on the protein synthetic capacity of cells. While both the p70S6K and 4EBP arms of the mTORC1 pathway have been implicated in the regulation of TOP mRNAs [15], other studies using both inhibitors and knock-out cell lines indicate that the 4EBPs are responsible for the regulation of these subsets of mRNAs [14,16]. Thus, the 4EBP-mediated translation regulation largely determines the protein synthetic capacity of a cell.

While the mTORC1 pathway and the role it plays in the activation and repression of protein synthesis in response to external stimuli, such as nutrients, mitogens and stress, are understood in detail, its role in fine-tuning the levels of protein synthesis in quiescent cells is not well understood. We have previously shown that resting state B and T lymphocytes, which exist in a quiescent state, exhibit distinct metabolic profiles, which reflect the functional needs of the cell [17]. T cells exhibit increased uptake of energy sources such as glucose and fatty acids compared with B cells. In accordance, T cells exhibit higher ATP levels and mitochondrial activity to meet the energy demands of the higher mobility of these cells. On the other hand, B cells exhibit an increased uptake of HPG, an amino acid analog, indicating increased protein synthesis. The higher level of protein synthesis in the B cells compared with the T cells, despite both of them being quiescent cells, was intriguing. This system provided us the opportunity to address the mechanisms by which different types of cells under quiescence maintain different protein synthetic capacity.

The data presented here show that resting state B cells exhibit higher protein synthesis levels than T cells in vitro and in vivo. B cells maintain an intermediate level of active mTORC1 pathway that maintains higher protein synthesis levels. Importantly, the data uncovered a novel mechanism by which B cells uncouple the activated state of mTORC1 from the translational upregulation of TOP mRNAs by increasing the expression of 4EBP1 protein.

## 2. Results

### 2.1. B Cells Exhibit Increased Protein Synthesis Compared with T Cells In Vivo

Previously, we have shown increased uptake of the amino acid leucine and the methionine analog L-homopropargylglycine (HPG) in B cells compared with T cells, which was interpreted as higher protein synthesis [17]. To confirm this, we labeled nascent proteins in resting-state B and T cells ex vivo with S35-labeled methionine and cysteine. Indeed, the S35 incorporation was higher in the B cells compared with the T cells (Figure 1A). The rate of protein synthesis is sensitive to external stimuli, including nutrients, mitogens and the cell microenvironment, which are affected by the cell isolation process. To ensure that this difference in the protein synthesis levels between B and T cells is a normal physiological phenomenon, we analyzed the rate of protein synthesis in vivo using the SUnSET assay. Here, the incorporation of the antibiotic puromycin in the nascent proteins was monitored to estimate the rate of protein synthesis [18]. The puromycin incorporation in splenic B and T cells 30 min after intraperitoneal injection of puromycin was checked by Western blotting. The puromycin incorporation was higher in the B cells compared with the T cells in vivo, clearly showing that B cells exhibit increased protein synthesis in vivo as well (Figure 1B).

### 2.2. CD74 Is One of the Highly Translated mRNAs in B Cells

Metabolic labeling of nascent proteins with S35 (Figure 1B) revealed that a band with mobility above 30 kDa dominated the B cell nascent proteome, while it was absent in the T cells. An equivalent band at this molecular weight was absent in the Coomassie-stained gel, indicating that the protein(s) that contribute to this band may be short-lived. A pulse-chase analysis of S35 incorporation, where the cells were incubated with S35-labeled amino acids for 30 min, followed by a chase with unlabeled amino acids, shows that the intensity of this band (Figure 1C, asterisk) decreases faster than the intensity of a 43 kDa band, which is abundant in both an autoradiogram and the Coomassie-stained gel (Figure 1C, arrowhead). Moreover, the relative intensity of this band reduced after activation with LPS (Figure 1D). B cells are professional antigen-presenting cells expressing the MHC Class II molecules. CD74 is a molecular chaperone that helps the assembly of the MHC Class II molecules. During the MHC II complex assembly process, CD74 is degraded to form a peptide bound to the MHCII complex [19]. Since the migration of this band corresponds to the molecular weight of CD74, and it is degraded rapidly, as expected for CD74, we asked if this is indeed CD74. We analyzed the S35 incorporation pattern in B cells from WT and CD74KO mice and found this to be missing from the CD74KO B cells (Figure 1E). Interestingly, among macrophages and dendritic cells, which constitute other APCs, this band was observed only in the dendritic cells, indicating the differences in the dynamics of expression of CD74 in these cell types (Figure 1F). Since CD74 expression and the dynamics of its protein levels are integral parts of antigen presentation, it is interesting to know what are the other classes of genes that are highly translated in the B cells.

### 2.3. Genes Associated with Antigen Presentation Are Highly Translated in B Cells

To further understand the nature of genes that are highly translated in B and T cells, we analyzed our previously published ribosome profiling data [17], where ribosome-bound mRNAs were detected and quantified at the global level. First, we identified the mRNAs that are translated at very high levels in each cell type. We defined the mRNAs that have a transcript per million (TPM) value higher than 1.5 times the inter-quartile range from the 3rd quartile of the TPM distribution to be highly translated in each cell type (Figure 2A). There were 256 mRNAs in the B cells and 261 in the T cells; 172 of which are common in both the B and T cells (Figure 2B; *p* value < 5.858 × 10^−279^). A gene ontology analysis of the 172 mRNAs that are highly abundant in both the cell types shows enrichment for protein synthesis-associated mRNAs, including ribosomal protein coding and translation initiation associated mRNAs (Figure 2C). This is not surprising, as protein synthesis-associated genes are well established to be highly abundant in most cells. On the other hand, 89 genes that are among the highly translated mRNAs only in T cells are enriched for TCR signaling, as well as protein synthesis-associated terms (Figure 2D). On the other hand, in support of our previous observation, the mRNAs that are highly abundant specifically in B cells are enriched for antigen processing and presentation-associated mRNAs (Figure 2E). Interestingly, among the mRNAs that are upregulated in B cells compared with T cells, the most abundant ones are the mRNAs coding for CD74 and other MHC Class II molecules (Figure 2F). Taken together, our data show that the antigen presentation-associated mRNAs form one of the most translated groups of mRNAs in the B cells.

### 2.4. mTORC1 Pathway Is Responsible for the Increased Translation in B Cells

mTORC1 is one of the crucial signaling complexes that play a central role in the regulation of translation. When we checked the levels of phospho-P70S6K and phospho-S6 as a proxy for mTORC1 activity, we found that their levels were higher in B cells compared to T cells (Figure 3A), suggesting upregulation of the mTORC1 pathway. Next, we asked if the upregulation of mTORC1 activity in B cells is responsible for higher protein synthesis in these cells. The previous flow cytometry-based quantification of pS6 in B and T cells did not show a significant increase [17], while the Western blot analysis shown here clearly indicates upregulation. Treatment with rapamycin, an inhibitor of the mTORC1 pathway, indeed led to a decrease in the phospho-S6 levels in B cells (Figure 3B), confirming that the increased pS6 levels observed in the B cells are indeed due to mTORC1 activity. Interestingly, consistent with the lower rate of protein synthesis in T cells, the p-S6 levels were lower in the T cells compared with the B cells, and these levels were unchanged upon treatment with rapamycin (Figure 3B), indicating that the p-S6 levels in the T cells represent the basal activity of the mTORC1 pathway. In contrast, B cells indeed exhibit an activated state of mTORC1. The treatment of B cells with rapamycin led to decreased protein synthesis in the B cells, as measured by the HPG incorporation assay (Figure 3C,E), confirming that B cell protein synthesis is dependent on the activation of the mTORC1 pathway. Analogous to the levels of p-S6 in the rapamycin-treated T cells, the protein synthesis in the T cells was unaffected by the rapamycin treatment (Figure 3D,E). It is well established that, upon activation, both B and T cells upregulate the mTORC1 pathway and protein synthesis. In accordance with this, the activation of B cells with LPS led to a further increase in mTORC1 activity (Figure 3F). These results indeed show that, in the resting state, B cells exhibit an intermediate level of activation, which is downregulated by rapamycin treatment and further upregulated by activation.

Interestingly, we observed that the 4EBP1 protein levels were higher in the B cells compared with the T cells (Figure 3G). Mirroring the p-S6 levels, the B cells exhibited higher levels of 4EBP1 phosphorylation at Thr 37/46, as well. The phosphorylation of 4EBP by mTORC1 occurs in two stages. In the first stage, the residues of Thr 37/46 are phosphorylated; these sites have a very high affinity for mTORC1-mediated phosphorylation and are resistant to rapamycin treatment and serum deprivation [14,20]. The second stage of phosphorylation occurs on the residues of Ser 65 and Thr 70. These exhibit a lower affinity for mTORC1 compared to the first stage residues and are sensitive to a lower concentration of rapamycin [21]. These different phosphorylations have distinct effects on the function of 4EBP: Ser 65 and Thr 70 phosphorylations are essential for preventing the interaction between 4EBP and eIF4E, while Thr 37/46 are required for Ser 65/Thr70 phosphorylations [22]. Therefore, increased Thr 37/46 phosphorylation may maintain the 4EBP1 protein in a poised state by the higher mTORC1 in the B cells, but its repressive function depends on the Ser 65 and Thr 70 phosphorylations. Intriguingly, there was no difference in the Ser 65 phosphorylation levels between these two cell types (Figure 3G). The 4EBP1 mRNAs were also enriched in both the transcriptome and translatome of the B cells, indicating its transcriptional upregulation in B cells (Figure 3H). Given that the 4EBP total protein level is high in the B cells, these results would indicate that B cells have higher levels of 4EBP1 hypo-phosphorylated at Ser 65, thereby muting the effect of increased mTORC1 on the 4EBP targets.

### 2.5. Both B and T Cells Exhibit Reduced Translation Efficiency of TOP mRNAs

Previous studies have shown that the regulation of the translation efficiency of mRNAs is primarily performed by 4EBPs. The data presented here indicate that upregulation of 4EBP1 protein levels in B cells could circumvent the activated state of mTORC1. If this is true, despite having a higher mTORC1 activity, B cells may not exhibit increased translation efficiency (the fraction of mRNAs that are ribosome-bound out of the total mRNAs available). To address this, we performed a combined analysis of Ribo-Seq and RNAseq data from the resting B and T cells to calculate the translation efficiencies of the mRNAs.

A differential analysis of translation efficiencies in B and T cells shows that a lesser number of genes exhibit a higher translation efficiency in the B cells compared with the T cells (Figure 4A). The mRNAs coding for MHC Class II proteins, as well as CD74, are not among the genes that show increased translation efficiency in the B cells. Intriguingly, an analysis of TPM values obtained from the Ribo-Seq experiment reveals that these mRNAs, which show increased TE in the B cells, are not very abundant in the translatome and are highly unlikely to be contributing towards the increased protein synthesis that we observed using the S35 and puromycin incorporation assays (Figure 4B). A comparison of the abundance of the differentially expressed mRNAs identified from the RNAseq of the B and T cells shows that the CD74 and other MHC Class II mRNAs are among the most abundant ones in the transcriptome of B cells, suggesting that the high levels of translation of these mRNAs are driven by the abundance of the mRNAs rather than by preferential translation (Figure 4C).

Since TOP mRNAs are considered bona fide targets of mTORC1 through the 4EBP–eIF4E axis, we compared the translation efficiency of these mRNAs in B and T cells (Figure 4D,E). As predicted by the increased expression levels of 4EBP1 in B cells, the TE of the TOP mRNAs was dramatically lower than the other mRNAs in B cells as well as in T cells. Moreover, a comparison of the translation efficiencies of different classes of mRNAs associated with protein synthesis also shows that they have similar TE in B and T cells (Figure 4F). Thus, our data show that B cells have a higher rate of protein synthesis driven by mTORC1 activity. However, the bona fide targets of mTORC1, the TOP mRNAs, are still translated at a lower level, most likely due to an upregulation of 4EBP1 expression.

### 2.6. Amino Acids Are Essential for Higher mTORC1 Activity in B Cells

Next, we asked what are the signals that maintain the intermediate levels of activation of mTORC1 in B cells. The mTORC1 pathway is activated by a variety of signals including mitogens and nutrients, especially amino acids. AKT is an established activator of mTORC1, which connects several receptor-mediated pathways to mTORC1 [23]. The activation of mTORC1 by amino acids is also known to be associated with AKT activation [24]. We checked two activation marks of AKT, phosphorylated Ser 473 (performed by mTORC2) and Thr 308 (performed by PDK1), and found that both are higher in B cells (Figure 5A,B). Thr 308 is performed by PDK1, which is activated by several cell surface receptors through PI3K. On the other hand, AMPK, a well-known repressor of mTORC1, is unchanged in B and T cells (Figure 5C). Our experiments on the amino acid uptake by B cells have shown that there is indeed higher amino acid uptake by B cells [17]. To further understand the stimulants that activate the mTORC1 complex in B cells, we washed these cells with PBS to deplete the nutrients and serum, followed by culturing for 1 or 2 h in complete RPMI medium. Upon nutrient and serum depletion, the mTORC1 activity was drastically reduced in the B cells (Figure 5D). After transferring these cells to the complete medium, the mTORC1 activity was upregulated within 1 h (Figure 5D). To further delineate if the stimulants are from the serum or nutrients in the medium, we either depleted amino acids or serum from the media during incubation. When the cells were grown in a medium that lacked three amino acids, Leu, Arg and Lys, the mTORC1 activity was down to the basal level, and replenishing these amino acids restored the mTORC1 activity (Figure 5E), clearly indicating that amino acids are essential for the maintenance of mTORC1 activity in these cells. On the other hand, the depletion of serum alone did not affect the mTORC1 activity in these cells (Figure 5F). Taken together, these cells indicate that B cells maintain increased levels of the mTORC1 pathway to achieve higher levels of protein synthesis.

## 3. Discussion

How different cell types maintain their steady state protein synthesis and metabolism to meet their specific demands is not well understood. The data presented here show that, during the resting state, B and T lymphocytes exhibit differential protein synthetic capacity and this difference could be functionally important, as it is mostly driven by the mRNAs that are coding for proteins associated with antigen presentation. B cells, along with dendritic cells and macrophages, constitute the major professional antigen-presenting cells in mammals [25]. Upon recognition of antigens by B cell receptors, these receptor–antigen complexes are internalized, processed and presented on MHC Class molecules for recognition by helper T cells [26]. This interaction between B and T cells is essential for the completion of the B cell activation process to form antibody-secreting cells. Our previous analysis has shown that, as expected, the genes associated with antigen processing and presentation are upregulated in the translatome of B cells compared to T cells [17]. The observation made in this study shows that mRNAs coding for MHC Class II and their chaperone CD74 are among the most abundant mRNAs in both the transcriptome and translatome of B cells, reiterating that the B cells, at the resting state, are geared for antigen presentation. Interestingly, the intensity of the S35-labeled band, corresponding to CD74, was high in dendritic cells, as well, while it was not as prominent in macrophages; detailed analysis is required to characterize the translatome of these cell types. Furthermore, future studies are required to establish if such changes in the rate of synthesis of antigen presentation-associated factors indeed affect the kinetics of antigen presentation in these cells.

A novel concept brought about by this study is the differential regulation of different arms of the mTORC1 pathway to maintain optimal levels of protein synthesis. The translational repressor 4EBP1 binds to eIF4E and represses cap-dependent translation initiation. Importantly, mRNAs that contain the terminal oligopyrimidine tract (TOP mRNAs), which are constituted mostly by ribosomal protein-coding genes and translation factors, are preferentially inhibited by the 4EBPs [13]. This inhibition is expected to maintain lower ribosomal protein levels, thereby keeping protein synthetic capacity of cells under check. Cell growth and proliferation require rapid upregulation of ribosomal proteins and the inhibition of 4EBPs by the mTORC1 pathway is essential to achieve this. There is evidence that shows differential effects of 4EBPs and p70S6K on cell growth and proliferation in different cell types [9,10,12]. However, the present study identifies a mechanism that resting B cells use to regulate protein synthesis without affecting overall cell growth or division. Here, the upregulation of the 4EBP1 protein offsets the activated state of mTORC1, thereby maintaining lower translation efficiency for the TOP mRNAs. The differential sensitivity of different 4EBP1 sites to mTORC1 plays an important role in this regulation. The highly sensitive sites Thr 37/46, which act as a priming site, are phosphorylated at a higher level in B cells, while the de-repression-effector site Ser 65 is not. This could be because the Ser 65 phosphorylation requires higher levels of mTORC1 activity. Thus, in effect, the 4EBP1 is expressed at a higher level and is primed in the B cells but is not fully functional, maintaining the low translation efficiency for the TOP mRNAs.

Previous work has shown that B cells exhibit higher amino acid uptake compared to T cells, and the data presented here show that amino acids are essential to maintain the increased mTORC1 activity in these cells. mTORC1 is known to be activated by signaling through the PI3K-AKT pathway, as well. The increased phosphorylation of AKT indicates that, probably, the cell surface receptor-mediated signaling could also contribute to the activation of the mTORC1 pathway. The serum deprivation experiment, however, shows that the serum-derived factors may not be responsible for the activation of AKT. This raises the possibility that the autocrine function of a B cell factor could be responsible for maintaining the mTORC1 activity in B cells. Further studies are required to fully elucidate the mechanisms that lead to activation of mTORC1 in B cells and how 4EBP1 is upregulated in these cells.

## 4. Materials and Methods

### 4.1. Animal Maintenance

Different strains of mice, C57BL/6 (B6) and CD74 KO, were obtained from the Jackson Laboratory (Bar Harbor, ME) and maintained in the animal facility of the National Institute of Immunology. All the mice used in the experiments were 6–8 weeks old. The study was carried out in accordance with the guidelines and recommendations of the Institutional Animal Ethics Committee.

### 4.2. Cell Isolation

The B and T cells were MACS-sorted using Miltenyi microbeads. B220 beads were used for B cell sorting, while CD90.2 beads were used to positively select T cells. The sorting was done as per the manufacturer’s instructions. The splenocytes were isolated and staining was performed at 4 °C for 30 min, followed by washing the cells with PBS. The cells were resuspended in complete RPMI (cRPMI) medium and sorted. The sorted cells were incubated in cRPMI for 2 h at 37 °C.

### 4.3. RNAseq and Ribo-Seq

For the RNAseq experiment, RNA was isolated from splenic B and T cells and libraries were prepared using the TruSeq RNA library prep kit v2. The Ribo-Seq experiment was described previously [17]. The differential analysis was performed using DEseq2 [27]. All the other analyses were performed using R and python scripts developed in-house, which are available upon request.

### 4.4. S35 Labeling of Proteins

The B220 and CD90.2 positive cells were MACS-isolated using microbeads (Miltenyi biotech 130-121-278 (CD90.2), 130-049-501(B220)). The cells were counted and 10 mn cells were kept in methionine-free medium for 30 min, followed by a 2 h pulse of S35-labeled methionine and cysteine mix. After 2 h, the cells were centrifuged and washed with PBS twice. For the chase experiments, unlabeled amino acids were added in excess and incubated for different time periods. The cells were lysed using RIPA buffer, followed by quantitation using BCA. An amount of 20 ug of total lysate was loaded on a 12% SDS gel for both the B220 and CD90.2 positive cells. The gel was stained using CBB and dried for 90 min. The gel was exposed to a phosphorimager screen overnight, and the scanning was done using the Typhoon scanner.

### 4.5. Western Blotting

The B220 and CD90.2 positive cells were sorted using Miltenyi microbeads by positive selection. The cells were counted and kept in complete RPMI medium (10 mn/mL) for 2 h. The cells were then washed twice with ice cold PBS and lysed in RIPA buffer containing a protease inhibitor cocktail and phosSTOP phosphatase inhibitors. The lysate was quantified using BCA, and 40 ug protein was loaded onto an SDS gel. The proteins were transferred onto a nitrocellulose membrane and blocked with 2.5% BSA (Merck) for 1 h at room temperature, followed by probing with primary antibody overnight at 4 °C. The blot was washed thrice with TBST before incubating with a secondary antibody for 1 h at room temperature. The blot was developed using the LAS500. The antibodies are listed in Table 1.

### 4.6. Puromycin Incorporation Assay

For the in vivo SUnSET assay, an intraperitoneal injection of puromycin of 0.040 μmoles/gram in PBS was given to the mouse and, post injection, after 30 min, the spleen was harvested, and the B220 and CD90.2 positive cells were sorted from the same mouse. The cells were lysed using RIPA buffer and 40 ug protein was loaded onto an SDS gel. The gel was transferred onto a nitrocellulose membrane and stained with Ponceau for loading control. The immunoblotting was performed using an anti-puromycin antibody.

### 4.7. HPG Incorporation Assay

The ex vivo splenocytes were kept in a methionine-free medium (vehicle or inhibitors) for 45 min at 37 °C, followed by the addition of 25 nM HPG for 2 h at 37 °C. After 2-h, the incubation cells were washed with PBS. The cells were then stained for surface markers by incubating with surface antibodies for 45 min at 4 °C in PBS. The cells were then again washed with PBS and fixed using 4% paraformaldehyde (PFA) for 10 min at room temperature. The cells were then washed with PBS containing 1% BSA and permeabilized using 0.2% Triton X 100 for 30 min at 4 °C. Next, the cells were washed with PBS containing 1% BSA, followed by incubation with freshly prepared Click-It reaction mix for 30 min at 37 °C in the dark. The cells were then washed with PBS containing 1% BSA and analyzed using FACSVerse. The Click-It reaction components included TRIS (100 mM), CuSO4 (1 mM) (Qualigens, Mumbai, India), L-ascorbic acid (20 mM; BioBasic, Markham, ON, Canada) and Alexa-Fluor488 azide (20 μM; Molecular Probes, Invitrogen). For the marginal zone and follicular B cell HPG assay, the cells were sorted first using FACS, and then the HPG assay was performed as mentioned above.

## 5. Conclusions

Translation activation and repression in response to growth factors or stress is relatively well understood. However, how different cell types, such as resting lymphocytes, muscle stem cells, etc., which remain in quiescent state for most of their life, maintain distinct levels of basal protein synthesis was not known. This work highlights the difference in the overall protein synthesis levels between resting B cells and T cells. T cells appear to exhibit a truly basal level of translation (without any aid from the mTORC1 pathway), while the increased levels of translation in B cells is due to an intermediate level of activation of mTORC1. Moreover, the upregulation of 4EBP1 protein in B cells maintains the TOP mRNA translation at a lower level, probably to avoid the unwanted upregulation of ribosome biogenesis.

## Figures and Tables

**Figure 1 ijms-23-16017-f001:**
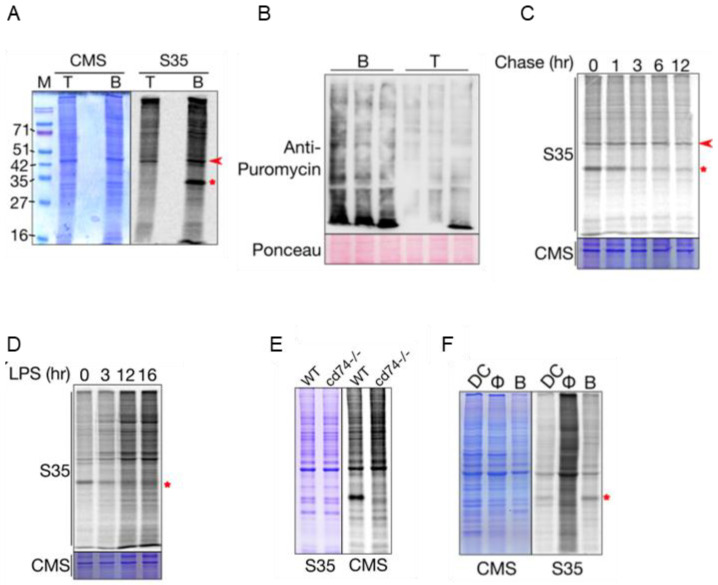
B cells exhibit higher protein synthesis compared with T cells. (**A**) S35 incorporation assay for estimating protein synthesis in B and T cells ex vivo. Cells were incubated with S35-labeled methionine/cysteine mixture for 30 min, followed by the analysis of cell lysates using SDS-PAGE. Coomassie-stained gel serves as the loading control. (**B**) Puromycin incorporation assay to estimate protein synthesis in B and T cells in vivo. Cells were isolated 30 min after puromycin injection and analyzed using Western blotting. Three lanes for each cell type represents data from three biological replicates. Ponceau-stained membrane serves as loading control (**C**) Pulse-chase assay to show the change in the intensity of the 33 kDa band (*). Another band above 42 kDa (arrow) does not show difference in intensity with time. Coomassie-stained gel serves as the loading control. (**D**) S35 incorporation assay of B cells activated with LPS. (**E**) S35 incorporation assay in WT and CD74KO mouse B cells. (**F**) S35 incorporation assay in different antigen-presenting cells-dendritic cells (DC), macrophages (Φ) and B cells (B). CMS = Coomassie-stained gel.

**Figure 2 ijms-23-16017-f002:**
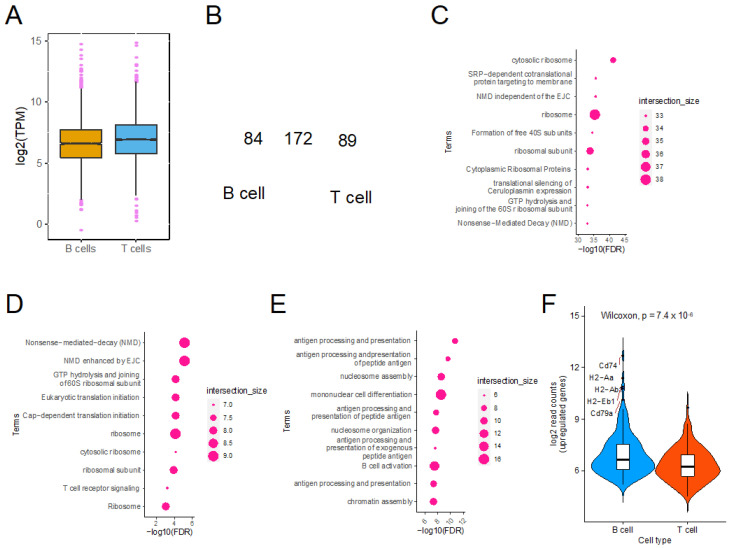
Antigen presentation-associated genes are among the top translated mRNAs in B cells. (**A**) Boxplots showing the distribution of transcript per million (TPM) values of resting state B and T cell Ribo-Seq data. (**B**) A Venn diagram showing the overlap between highly translated genes in B and T cells. Hypergeometric test *p*-value < 5.858 × 10279. (**C**–**E**) plots showing the pathways/GO terms enriched in the genes that are highly translated in both B and T cells (**C**), only in T cells (**D**), and only in B cells (**E**). (**F**) A violin plot showing the read distribution of genes upregulated in B and T cells in the Ribo-Seq data.

**Figure 3 ijms-23-16017-f003:**
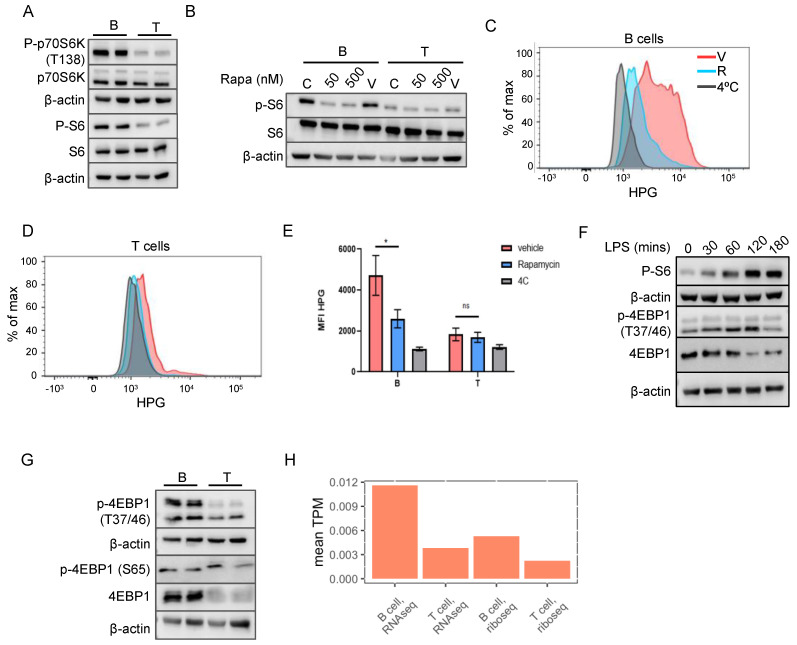
mTORC1 pathway activates protein synthesis in B cells. (**A**) Western blot analysis of different mTORC1 pathway intermediates in B and T cells. Two biological replicates per cell type are shown. (**B**) Western blot analysis of phospho-S6 in B and T cells treated with rapamycin. Lane labeled C indicates control lane without any reagents and the lane denoted as V shows vehicle control. (**C**,**D**) Density plots of HPG incorporation by B (**C**) and T (**D**) cells. (**E**) Quantitation of HPG incorporation of B and T cells under different conditions. * indicates *p*-value < 0.05. (**F**) Analysis of the activation of the mTORC1 pathway after LPS treatment of B cells. (**G**) Western blot analysis of 4EBP total protein as well as the phosphorylated forms in B and T cells. Two lanes per cell type indicate biological replicates. (**H**) Mean TPM of 4EBP in B and T cell Ribo-Seq and RNAseq data.

**Figure 4 ijms-23-16017-f004:**
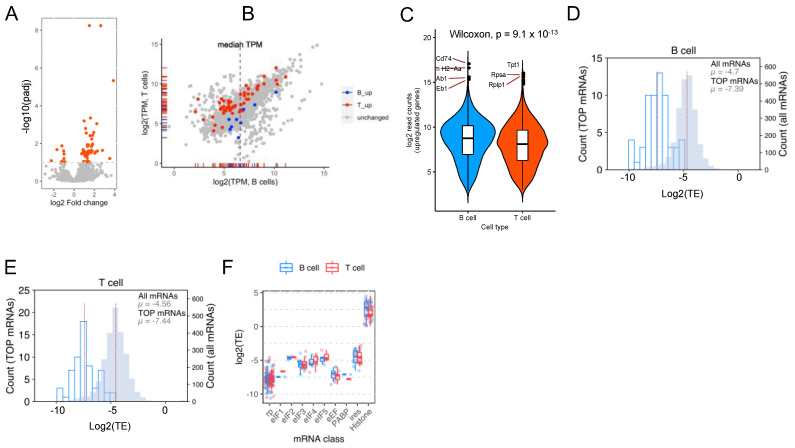
TOP mRNAs exhibit reduced translation efficiency in both B and T cells. (**A**) A volcano plot showing the results of a differential translation efficiency analysis of B and T cells. (**B**) A scatterplot for the log TPM values of B and T cell Ribo-Seq analysis, with the genes that show increased TE in B (blue) and T (red) cells marked. (**C**) A violin plot showing the distribution of mRNAs in B and T cell transcriptomes. Very highly expressed mRNAs are labeled. (**D**,**E**) Histograms that show expression levels of all genes (filled bars) and TOP mRNAs (open bars) in B (**D**) and T (**E**) cells. The mean log2(TE) of these classes of mRNAs is provided in the plot. (**F**) Boxplots comparing the translation efficiency of different classes of genes in the translatome of B and T cells.

**Figure 5 ijms-23-16017-f005:**
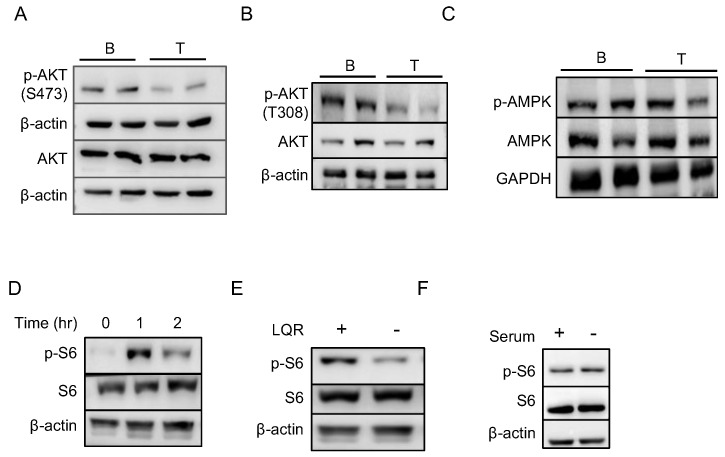
Nutrients as well as signaling pathways control mTORC1 pathway in B cells. (**A**,**B**) Phosphorylation of AKT at positions S473 and T308. (**C**) Phospho-AMPK levels in B and T cells. In A, B and C, biological replicates for each cell type are shown. (**D**) Representative Western blots of p-S6 levels in B cells immediately after isolation and at different time points after culturing in RPMI medium. (**E**) p-S6 levels with and without amino acids leucine, glutamine and arginine. (**F**) phospho-S6 levels after serum starvation in B cells.

**Table 1 ijms-23-16017-t001:** List of antibodies used in this study.

Antibody	Make	Catalogue Number
S6	CST	2217S
p-S6 (Ser 240/244)	CST	2215S
4EBP1	CST	9452
p-4EBP1 (T37/46)	CST	2855
p-4EBP1 (S65)	CST	9451S
p70S6K	CST	9202
p-P70S6K (T389)	CST	9205
AMPK	CST	5831T
p-AMPK (T172)	CST	40H9
AKT	CST	9272
p-AKT (S473)	CST	9271
p-AKT (T308)	CST	4056
GAPDH	CST	2118
Beta actin	CST	4967S
anti-puromycin	Sigma-Aldrich	32160702

## Data Availability

Publicly available datasets were analyzed in this study. This data can be found here: [GSE151718].

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
