# Peer review of "Differential Regulation of Two Arms of mTORC1 Pathway Fine-Tunes Global Protein Synthesis in Resting B Lymphocytes"

_ijms, 2022, doi:10.3390/ijms232416017_

Round 1

Reviewer 1 Report

There is very little data here that would confirm the hypothesis about the difference between B and T lymphocytes, although the hypothesis is not entirely clear.

Why are B and T lymphocytes compared at all when they always have different functions in the immune system?

In my opinion, it would be much more interesting to compare the synthesis of proteins and some other molecules during the differentiation of B lymphocytes, then during the transformation into plasma cells, which would show the mechanism, kinetics and function.

Thus, we do not know in which phase individual T and B lymphocytes were, nor was the phase of the cycle examined; dormant or activated

Reviewer 2 Report

This is an interesting study comparing protein synthesis in T and B cells and the potential cross talk between 4EBP1 and S6K in regulating protein synthesis in B cells.  

The manuscript could be improved by:

1) listing catalog numbers of the actual antibodies used

2) better quality of images in Figs 2 and 4; more detailed and careful labeling of figure legends and figures

3) Fig 1A --do the authors mean T389?

4) Fig 1B--which phospho site of S6 is shown? Ser 240/244?

5) Fig 5F -- it would have been nice to see a follow up experiment +/- amino acid add back..

6) Lines 202-205: The relative activity of mTOR is context dependent and nutrient dependent... Under what conditions were the cells harvested for the western blot? Would it be worth starving the cells of amino acids and reactivate with amino acids + 10% FBS...  Is it possible that 4EBP1 may not show differences if the cells are primed to respond to nutrient signaling?

Reviewer 3 Report

In this paper, the authors have performed a comparative analysis of the translatome and transcriptome of B cells and T cells by Riboseq and RNAseq, respectively. The authors show that high mTORC1 activity leads to reduced translation of TOP mRNAs. Although the study is interesting, there are some points that could be addressed.

In figure 1, the authors could show the number of genes that enrich a specific pathway (how many genes contribute) to the pathway. It is surprising that all the circles look the same size, as I know that different sizes of the genes would contribute to enrichment.

The significance of the overlap could be shown.

For figure1, what kind of T cells are used- CD4, CD8, or general? How do the authors know these T cells are not activated?

Did the authors check CD74 KO mice B cells for AKT or mTOR activation?

Is the cell cycle of B cells and T cells different?

Did the authors compare their data with any published dataset (Haemosphere) to check how the normal levels vary? Does it match their observations? How do the authors add knowledge to the current understanding of B cells, since immgen and more databases have already checked the normal expression of the genes in mice?

Did the authors validate any highly translatable mRNAs by qPCR or other orthogonal methods, such as western blot?

Could the authors activate B cells and T cells simultaneously, and can the author see similar differences in mTORC1?

Minor-

There are a couple of spelling mistakes in the legend. The authors could correct the mistakes.

Round 2

Reviewer 1 Report

I did not see that the authors changed the text in many ways in order to improve the work compared to the previous version

Author Response

We modified the text of the paper to correct grammatical and style problem. We have used professional software (Grammarly) to modify the text of the paper.